# Using Wikidata in the European Literary Bibliography: A Reproducible Approach

Gustavo Candela | University of Alicante | Spain | gcandela@ua.es
Cezary Rosiński | IBL PAN | Poland | cezary.rosinski@ibl.waw.pl
Arkadiusz Margraf | PSNC | Poland | margraf@man.poznan.pl

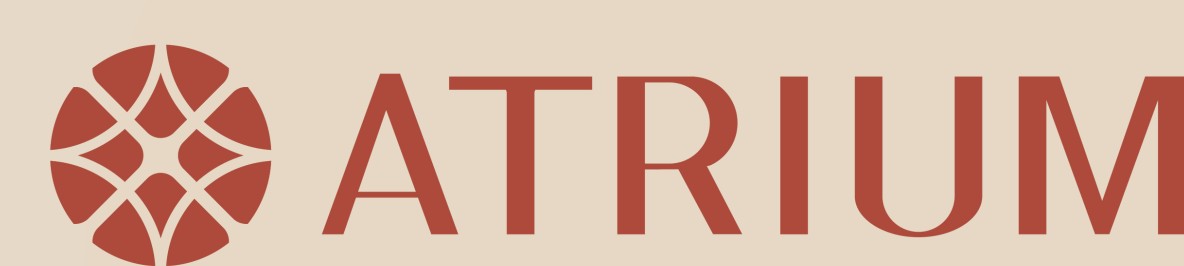

## INTRODUCTION

**GLAM institutions** (Galleries, Libraries, Archives, and Museums) are increasingly making their resources available as machine-readable "Collections as Data." They leverage diverse materials—texts, images, postcards, metadata, bibliographic databases, audio, video, maps—and modern techniques (network analysis, Named-Entity Recognition, Entity Linking) to facilitate reuse, including applications in NLP and Machine Learning. Initiatives such as digital Labs and "Collections as Data" projects, along with community groups like the Bibliographical Data Working Group within DARIAH, promote open licensing and collaboration between data curators and researchers.

### KEY GLAM AREAS
- Standardized documentation & checklists
- Semantic Web & Linked Open Data
- Bibliographic initiatives & data dumps
- Quality assessment & tooling

## BACKGROUND

The **European Literary Bibliography** (ELB, literarybibliography.eu) is an ecosystem for processing, integrating, enriching, presenting, and sharing open literary bibliographical datasets internationally. It is developed jointly by Czech Literary Bibliography and Polish Literary Bibliography. ELB is open to bibliographers, data curators, data experts, and lay users, and intends to enhance the understanding and creative exploration of the European literary landscape, promoting literary metadata integration and reuse.

The **ATRIUM project** (Advancing fronTier Research In the arts and hUManities, atrium-research.eu) offers **Transnational Access** (TNA) scheme. This initiative is designed to support Arts and Humanities researchers by providing access to expert knowledge, mentorship, and tools from leading Data Management organisations.

## FRAMEWORK

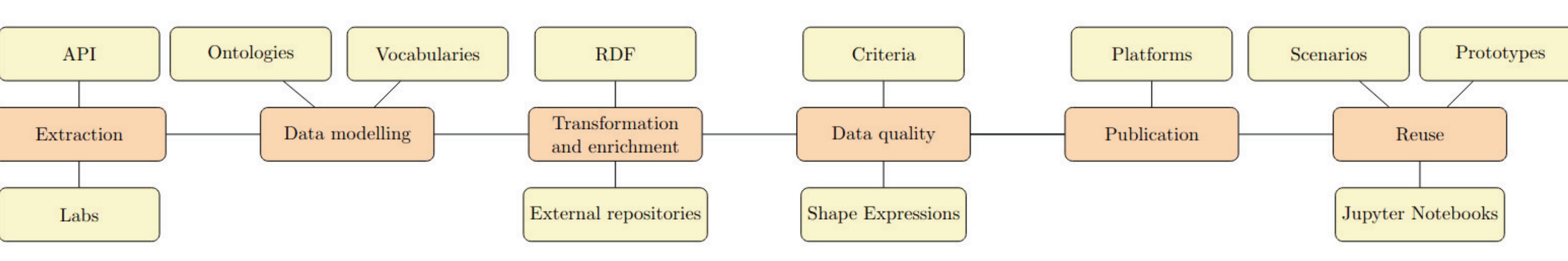

Figure 1. Framework employed to transform bibliographic metadata into Collections as Data.

**Data extraction** refers to the selection of data relevant to a specific topic (e.g., author, organization or theme). **Data modelling** aims at ensuring machine-readable bibliographic metadata, using ontologies and vocabularies. The **transformation and enrichment** step refers to the transformation of the data into Linked Open Data (LOD) using RDF to describe metadata as triples as well as the use of Wikidata to enrich the metadata. The **data quality** step ensures the high quality of the RDF data. The **publication** requires the inclusion of additional documentation including aspects such as provenance and licensing. Finally, the published datasets can be **reused** in various ways (e.g., data processing via Jupyter Notebooks and prototypes or research scenarios defined by DH scholars).

## COLLECTIONS AS DATA

Collections as Data is an initiative that promotes the **publication of digital collections** in a manner suitable for responsible **computational use**.

Within the Collections as Data framework, content of **cultural heritage institutions** is made available as a dataset structured to address a specific topic or research question including content in various formats and designed to facilitate computational analysis using digital methods.

Key aspects include:
- Providing a clear **licence and terms of use**
- Supplying rich **documentation or metadata describing the content and dataset** with provenance, selection criteria, and methodology
- **Structuring the dataset** to facilitate its reuse
- Making collection **accessible via API or data dumps**
- Using **machine-readable vocabularies** to describe the content and the dataset itself
- Providing **examples of use** (e.g., via Jupyter Notebooks)
- Suggesting a **citation format for the dataset** to promote its access, findability, reusability, and give credit to the institution

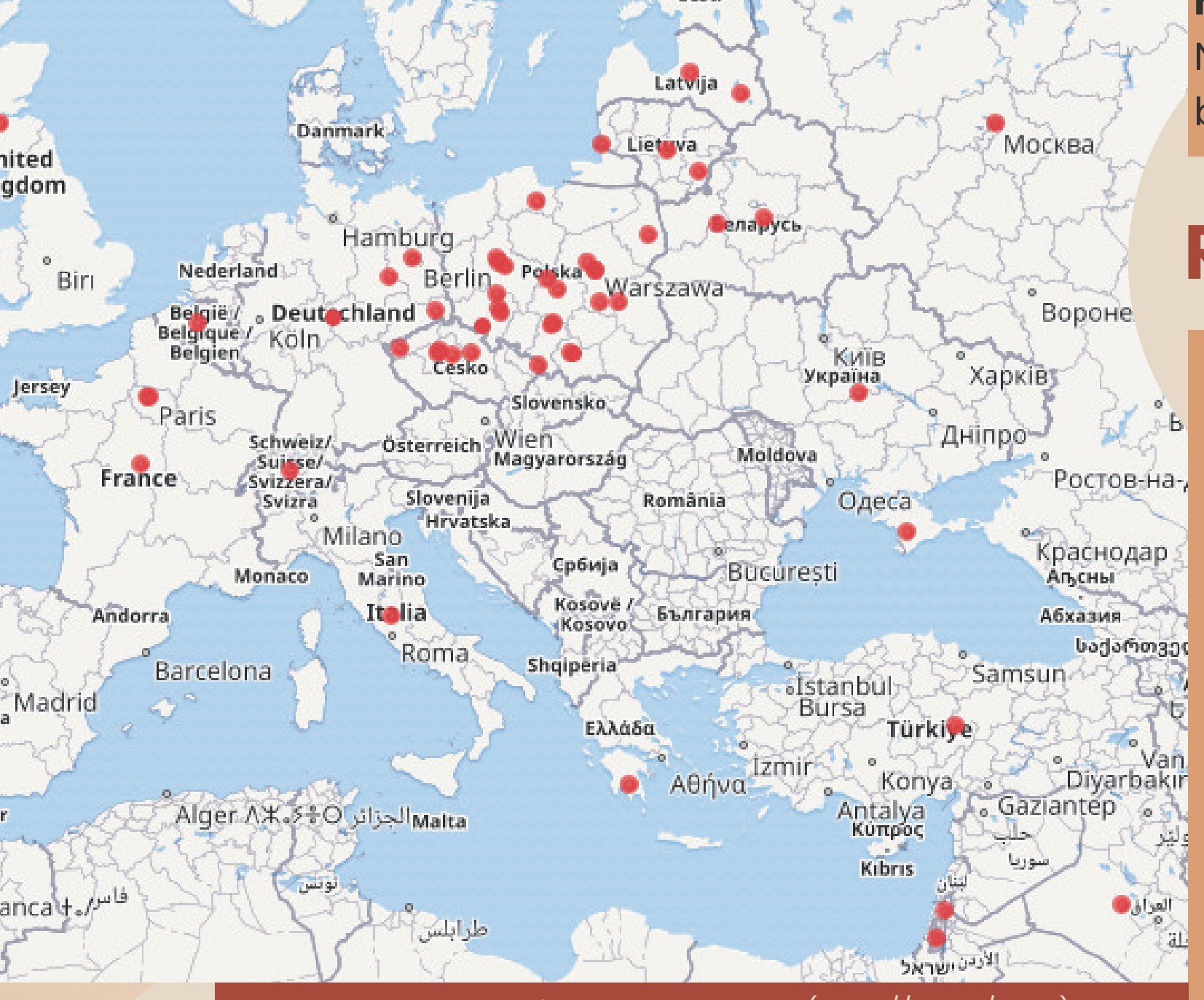

Figure 2. A map resulting from a SPARQL query (https://w.wiki/EFwU) related to places (births, deaths, publications) from the Adam Mickiewicz dataset.

## RESEARCH SCENARIOS

The ELB exemplifies how **GLAM data** can be integrated with **scholarly communication**. The research scenarios illustrate how the proposed framework can be applied to address specific challenges faced by GLAM institutions when their data are used for **scientific purposes**. Scenarios focus on **linking bibliographical resources with external authority databases**, such as Wikidata, to increase the possibility of inferring and viewing data. These are the 3 scenarios that we propose in this work:

1. Comparative analysis of provincial Spanish novels featuring vampires
2. Republican writers who migrated during the Spanish Civil War
3. Geographical distribution of publications dedicated to particular Spanish writers

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

## RESULTS

Our main contributions are:
1. A **framework** to publish and reuse Collections as Data based on bibliographic metadata
2. The **results** obtained using the proposed infrastructure, including **reproducible code**
3. A **selection of DH research scenarios** illustrating how the data could be explored and reused.

These contributions are intended to leverage the adoption of Collections as Data by GLAM institutions providing detailed documentation and code, as well as additional use cases based on bibliographic metadata.

| Dataset | Classes | Properties | No. triples | No. external links |
|---|---|---|---|---|
| Adam Mickiewicz | 7 | 19 | 47,631 | 863 |
| Miguel de Cervantes | 5 | 19 | 44,389 | 749 |
| National Library of Spain | 7 | 19 | 10,515,016 | 67,563 |

Table 1. Results obtained by applying the framework to selected datasets from the European Literary Bibliography

### VISIT OUR REPOSITORY

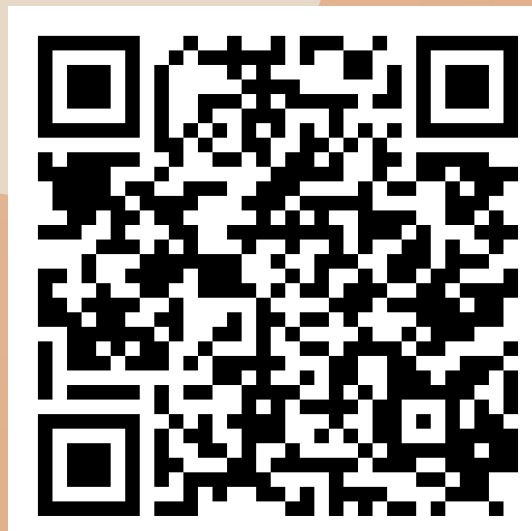

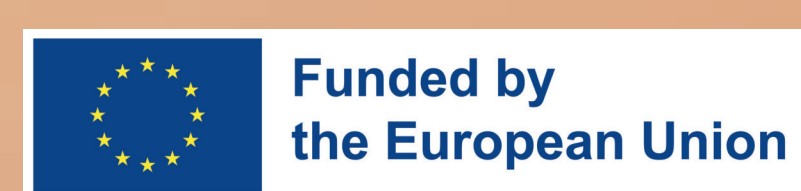

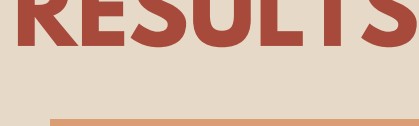

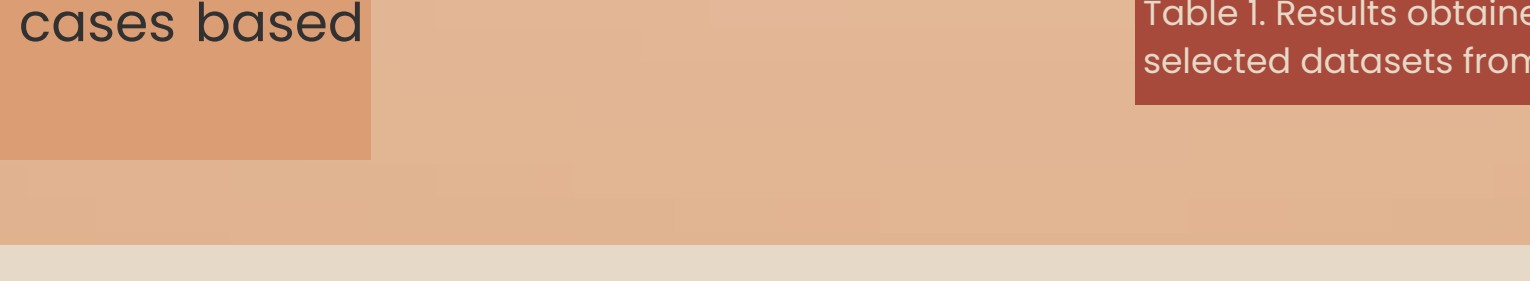

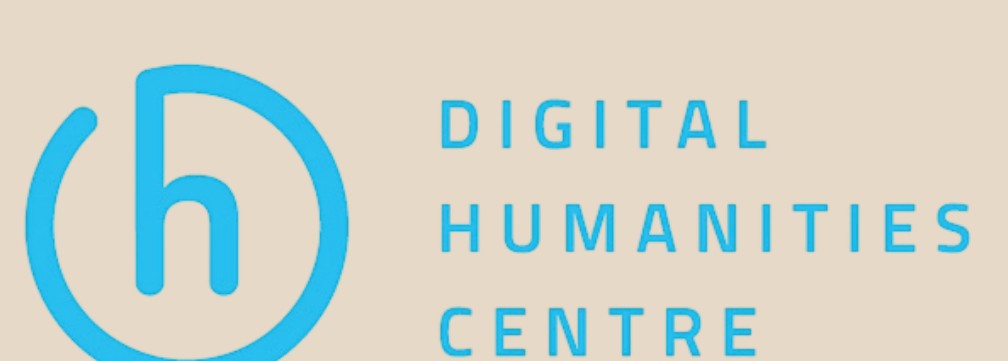

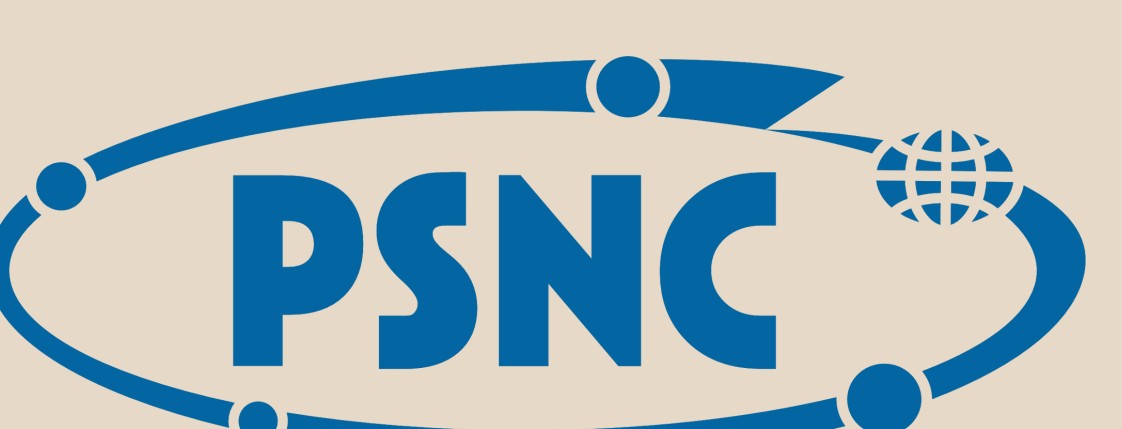

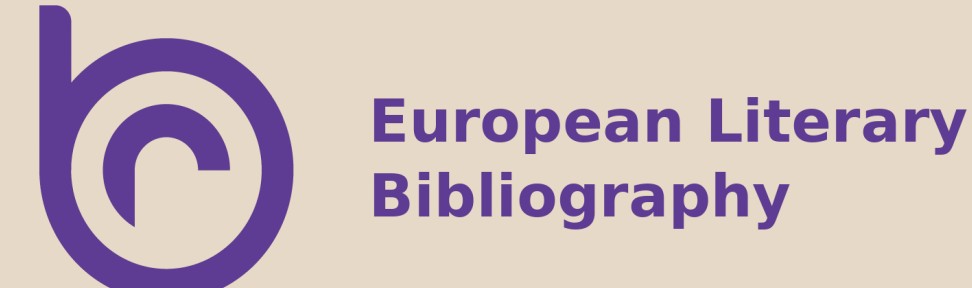

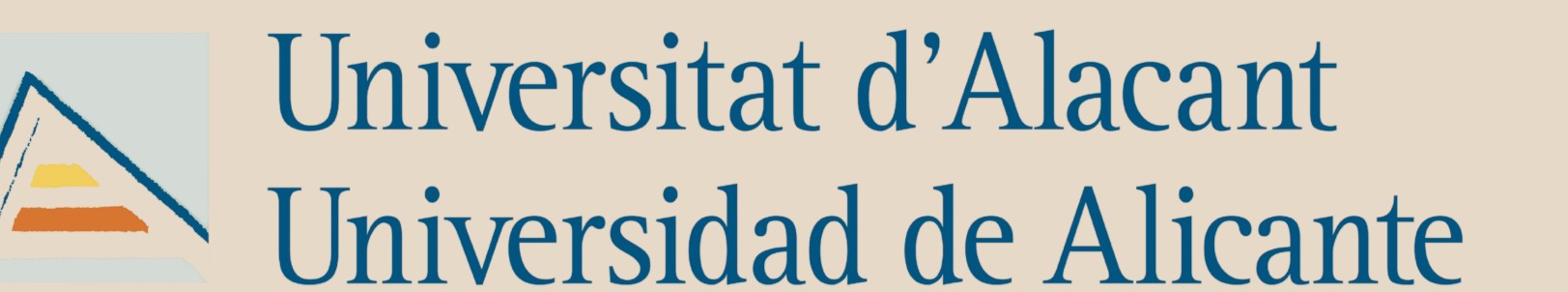