# OpenReview forum: "Using Wikidata in the European Literary Bibliography: A Reproducible Approach"
_wikimedia.it/Wikidata_and_Research/2025/Conference — WD&R Poster_

### Official Review · ~Annick_Farina1 · 2025-01-08
**very interesting research on the use of Wikidata in the European Literary Bibliography**

**Originality:** 5
**Impact:** 5
**Confidence:** 3

**Review:**

As part of dissemination of a research project (ELB) - Institute of Czech Literature (Czech Academy of Sciences) and Institute for Literary Research (Polish Academy of Sciences) the authors intend to present their work which could provide a reproducible framework including several steps for publishing and reusing digital collections based on literary bibliographies.

**Compliance:**

5

**Scientific Quality:**

5

---

### Official Review · ~Carlo_Bianchini1 · 2025-01-08
**A reproducible framework for publishing bibliographic data from European Literary Bibliography**

**Originality:** 5
**Impact:** 5
**Confidence:** 4

**Review:**

The proposal aims to present a reproducible framework for publishing and reusing collections as data, through data extraction, data modelling, transformation and enrichment, data quality control, publication, and reuse. It is a topic of high interest and the developed methodology is very valuable.

**Compliance:**

5

**Notes:**

I would strongly suggest to ask the author to change their presentation from a poster to a paper.

**Scientific Quality:**

5

---

### Decision · Program_Chairs · 2025-02-05

Accept (Poster)